# Effectiveness of conditional cash transfers (Afya credits incentive) to retain women in the continuum of care during pregnancy, birth and the postnatal period in Kenya: a cluster-randomised trial

Fedra Vanhuyse [1] , Oliver Stirrup,[2] Aloyce Odhiambo,[3] Tom Palmer,[2] Sarah Dickin,[1] Jolene Skordis,[2] Neha Batura [2] ,[2] Hassan Haghparast-Bidgoli [2] ,[2] Alex Mwaki,[3] Andrew Copas [2] [2]

[1]Stockholm Environment Institute, Stockholm, Sweden
[2]Institute for Global Health, University College London, London, UK
[3]Safe Water and AIDS Project, Kisumu, Kenya

**Correspondence to**
Dr Fedra Vanhuyse;
fedra.vanhuyse@sei.org

## ABSTRACT

**Objectives** Given high maternal and child mortality rates, we assessed the impact of conditional cash transfers (CCTs) to retain women in the continuum of care (antenatal care (ANC), delivery at facility, postnatal care (PNC) and child immunisation).

**Design** We conducted an unblinded 1:1 cluster-randomised controlled trial.

**Setting** 48 health facilities in Siaya County, Kenya were randomised. The trial ran from May 2017 to December 2019.

**Participants** 2922 women were recruited to the control and 2522 to the intervention arm.

**Interventions** An electronic system recorded attendance and triggered payments to the participant's mobile for the intervention arm (US$4.5), and phone credit for the control arm (US$0.5). Eligibility criteria were resident in the catchment area and access to a mobile phone.

**Primary outcomes** Primary outcomes were any ANC, delivery, any PNC between 4 and 12 months after delivery, childhood immunisation and referral attendance to other facilities for ANC or PNC. Given problems with the electronic system, primary outcomes were obtained from maternal clinic books if participants brought them to data extraction meetings (1257 (50%) of intervention and 1053 (36%) control arm participants). Attendance at referrals to other facilities is not reported because of limited data.

**Results** We found a significantly higher proportion of appointments attended for ANC (67% vs 60%, adjusted OR (aOR) 1.90; 95% CI 1.36 to 2.66) and child immunisation (88% vs 85%; aOR 1.74; 95% CI 1.10 to 2.77) in intervention than control arm. No intervention effect was seen considering delivery at the facility (90% vs 92%; aOR 0.58; 95% CI 0.25 to 1.33) and any PNC attendance (82% vs 81%; aOR 1.25; 95% CI 0.74 to 2.10) separately. The pooled OR across all attendance types was 1.64 (1.28 to 2.10).

**Conclusions** Demand-side financing incentives, such as CCTs, can improve attendance for appointments. However, attention needs to be paid to the technology, the

## Strengths and limitations of this study

► Technical issues with the electronic system and at times low participation of health workers resulted in many visits not being registered, and only 26% of the payments triggered automatically.

► Manual payments needed to be triggered, which resulted in delays.

► This delay in payment could have diluted the impact of the intervention and prevented service utilisation in the latter stages of pregnancy or after babies were born.

► As manual data abstraction from clinic registers and from the women's maternal clinic books was necessary, we obtained near complete data for antenatal care attendance but limited data on facility delivery, postnatal care and child immunisation and no data on referral attendance.

► The potential bias is however limited to a degree by our approach to analysis in which all outcomes are modelled simultaneously.

barriers that remain for delivery at facility and PNC visits and encouraging women to attend ANC visits within the recommended WHO timeframe.

**Trial registration** NCT03021070.

## INTRODUCTION

Every year an estimated 295 000 maternal deaths occur globally, with 99% occurring in low-income and middle-income countries, and almost two-thirds in Africa.[1] In sub-Sahara Africa, the maternal mortality rate was 542 deaths per 100 000 live births in 2017.[1] In Kenya, the maternal mortality rate in 2014 was 362 deaths per 100 000 live births,[2] categorised as 'high' according to the WHO.

Globally, an estimated 2.4 million infants die in the first month of life (40% of all deaths) and 1.6 million at age 1–11 months (25%).[3] Sub-Saharan Africa has the highest neonatal mortality rate at a median of 27 deaths per 1000 live births and the highest under-five mortality rate, with a median of 76 deaths per 1000 live births.[3] In Kenya, the median infant mortality rate was 21 deaths per 1000 live births and the median under-five mortality rate was 43 deaths per 1000 live births,[2] which are among the highest in the region.

Most maternal and neonatal deaths are avertible through the use of healthcare interventions that prevent or manage pregnancy-related complications such as postpartum haemorrhage and infectious diseases.[4 5] The importance of health service utilisation for maternal and child health outcomes has been extensively documented, including for antenatal care (ANC),[6–8] facility delivery,[9 10] postnatal care (PNC)[11] and across the continuum of care.[12] In Kenya, while 96% of women receive some form of ANC, less than three in five receive the four ANC visits that the WHO recommends.[2] Only one in five have their first ANC visit during the first trimester as recommended by the WHO,[2] which would allow to monitor the health of both mother and child more effectively. Almost four in ten babies are delivered at home, and 62% of newborns do not receive a PNC check-up in the first week after birth.[2] Almost half (47%) of mothers do not receive a PNC check-up in the first 2 days after birth.[2] Lower attendance at healthcare facilities could be due to a lack of skilled health workers, poverty, distance, lack of information, inadequate services and cultural practices.[13] Other research[14] reported costs charged for ANC visits, nurses' behaviour and the timing of the visits as the main barriers for attending ANC visits. In Kenya, a survey showed that financial barriers (costs of care for other children, food, new clothes), and lack of transport and distance to healthcare facilities were the main barriers.[15]

Aside from programmes to improve the quality and reach of the service (supply side interventions), demand side financing interventions have been set up to incentivise women to attend visits. Examples of such interventions include mobile phone text message reminders for ANC visits in Zanzibar[16] and for PNC visits in Tanzania,[17] as well as the use of conditional cash transfers (CCTs). In Kenya, a conditional cash transfer intervention in Vihiga County was found to increase facility delivery by 7.9 percentage points.[18] A recent study in Nigeria[19] found that payments for retention from ANC to PNC resulted in more women attending the visits (26% of women in the intervention arm compared with 12% in the control arm), leading to a 22% reduction in the stillbirth rate. Recent systematic reviews on the demand-side interventions for maternal care[20] and on CCTs[21–24] found increased utilisation of services, but not always better outcomes.

The Afya trial aims to test the effectiveness of a conditional cash transfer to retain women in the continuum of care, from their first ANC visit until their children reach 1 year of age in Siaya County, Kenya. The CCT aimed at tackling multiple barriers to care, as described in the trial's protocol[25]: women would receive equal-sized cash transfers following a visit (ANC, delivery at facility, PNC visit and childhood immunisation), as well as a reminder for their visit by text message and medical staff would be trained in the technology and incentivised for each woman they enrolled in the trial. Unlike a study in Nigeria,[19] in Afya, the CCT was done through a card reader system rather than cash and, additionally, it initiated transfers for each individual visit, rather than being conditional on receiving an entire package of care. In this paper, we present the results of the impact of the Afya trial.

## METHODS
### Study setting, design and randomisation
We conducted a cluster randomised controlled trial, with equal allocation to intervention and control arms, in Siaya county, Kenya. The units of randomisation were level 2 or 3 health facilities (Dispensaries and Health Centres, respectively). The randomisation of facilities was stratified by the six subcounties and ensured equal allocation to study arms within each stratum without any overlap of catchment areas, as described in detail in the trial protocol.[25] In summary, at a public forum with the county government early 2016, the implementing partner wrote the names of 60 shortlisted facilities on pieces of paper and folded them to hide the names, then included them in transparent boxes, one for each subcounty. Each subcounty had an (even) number of facilities to recruit to the trial proportional to subcounty size. The health management teams from each subcounty selected the pieces of paper, one by one. The first was allocated to intervention, second to control. For each selected facility, county officials from the selected subcounty mapped the location and catchment area of the facility on a large map of the county. If a subsequently selected facility had an overlapping catchment area with a previously selected facility, the newly selected facility was rejected, and another drawn to take its place. This process continued until 48 facilities were selected and allocated for the trial.

Health facility staff determined whether a pregnant woman met the study eligibility criteria by administering screening questions at the end of her first ANC visit, with the screening questions provided in the trial protocol.[25] All women meeting the criteria were eligible for recruitment during the study recruitment period. Criteria for enrolment were women attending their first ANC visit; long-term resident of the catchment area served by the health facility (living in the area for at least 6 months); access to a mobile phone that belongs either to themselves or to a member of their household or person whom they trust. The criterium on residence provided additional assurance that women went to the facility within their catchment area, thereby reducing contamination with other facilities. Oral informed consent was asked in

the local language, and then written down on the participant's enrolment form. Refusals were recorded.

## Intervention

The intervention was a CCT payment for each facility appointment attended for ANC, delivery, PNC and childhood immunisation. Detailed definitions can be found in online supplemental appendix S1. For each scheduled health visit made following enrolment, women in the intervention arm received a cash transfer of KSH 450 (US$4.5) on their mobile phones. Women at the control clinics were granted KSH 50 (US$0.5) mobile phone airtime for each scheduled visit to encourage them to bring their clinic booklet to appointments. In both trial arms, women were issued with a trial card at recruitment and at all facilities there was a card reader, which provided the connection between the trial card and an online portal which stored participants' data on visits and payments. Payments to the women were triggered by tapping the card on a card reader, which also logged the visit in an online portal.[26] In the event of problems with the card reader, or if the woman did not bring her card to the appointment, payments could alternatively be processed manually by contacting the implementing partner: once the visit was verified with the facility, the implementing partner entered the visit data in the portal, which would then trigger a payment as well nurses were given KSH 400 (US$4) per woman enrolled during the trial, and an additional KSH 100 (US$1) per woman enrolled at the end of the trial for their collaboration in the trial. These payments were transferred to the nurses electronically. Details of the intervention design are presented in the protocol.[25]

## Patient and public involvement statement

Patients or the public were not involved in the design, conduct, reporting or dissemination plans of the research. The assessment of the burden of the intervention on patients is reported in Dickin *et al*.[27]

## Trial outcomes

The primary outcomes were: (1) attendance or missed attendance at each eligible ANC appointment after recruitment; (2) delivery at a health facility; (3) attendance for at least one PNC appointment between 4 and 12 months after delivery; (4) attendance or missed attendance at each expected child immunisation appointment; (5) attendance at referrals to other facilities for ANC, PNC or child immunisation. We define 'eligible visits' for the purpose of statistical analysis of the impact of the intervention within a clearly defined framework, but all scheduled prenatal and postnatal clinic visits should have triggered a payment. The PNC attendance outcome described in the protocol was the 'proportion of required postnatal visits honoured after recruitment into the study'. However, we have used a simplified outcome here because the required appointment schedule was not recorded for each patient. We also restricted attendances to the period 4–12 months

after delivery because on blinded review of the available data, prior to writing the statistical analysis plan (SAP), very few visits prior to 4 months postdelivery were coded as PNC. The vast majority of visits prior to 4 months were recorded as vaccination appointments. though it is likely some women also received PNC. Vaccinations over the 12 months after delivery were recorded. The details of vaccinations given were not collected, and so the vaccination outcome is based purely on the number of recorded visits from an expected number of four.

The following secondary outcomes are also reported and analysed according to the trial arm: (1) attendance of all eligible maternal, newborn and child healthcare visits, both prenatal and postnatal, for each woman; (2) the count of attended ANC and child immunisation clinic visits eligible for the primary outcome variables for each woman; (3) the total number of ANC, child immunisation and PNC clinic visits (without applying any eligibility criteria) for each woman; (4) gestational age (GA) at first ANC visit (and enrolment to study). The secondary outcomes of clinic visit counts and GA at first ANC visit were not listed in the trial protocol but were prespecified in the SAP to aid interpretation of the study results.

The primary outcome of attendance at referrals to other facilities for ANC, PNC or child immunisation is not reported because of very limited data. The following planned secondary outcomes are reported without formal statistical analysis because of low levels of data completeness: maternal and neonatal mortality, self-rated wellness, exclusive breast feeding and contraceptive use. The following secondary outcomes were dropped because of lack of available data: timeliness of health visits (recorded visits could not consistently be matched up to scheduled dates), and infection screening. These changes were specified in the SAP. Online supplemental appendix S1 contains further details of changes made to the data collection and analysis compared with the protocol, arising from data collection challenges.

## Data collection

Data were collected throughout the trial from an electronic card reading system, and baseline and follow-up surveys. The electronic card reading system captured the enrolled women's phone number, expected delivery date, parity, the clinic she enrolled at, the visits she attended and the payments she received. The baseline survey, carried out by telephone following enrolment, collected sociodemographic data. We initially planned to conduct follow-up interviews with 50% of enrolled women at 6 and 12 months postdelivery to collect secondary outcomes. However, due to limited resources and lower than anticipated response rates after some 6-month interviews had been conducted, we adopted a pragmatic approach and conducted one follow-up survey at around 12 months after delivery.

Problems with the implementation of the technical system and periods of disengagement from clinic staff resulted in a large proportion of visits not being registered

on the system. Therefore, data on women's visits were manually extracted from clinic health records after the trial was completed. Trial staff visited each facility, after arranging for trial participants to be invited to attend the facility with their maternal clinic book. Data on the primary outcomes of ANC, child immunisation and PNC visits and delivery at a healthcare facility were extracted from maternal clinic books. The books very rarely contained records of any referral visits. For women who did not attend the data extraction, data on the primary outcome of ANC visits were extracted from health facility registers, but data on other visits were not available from this source. Any missing payments were transferred to the participating women following the manual data extraction at the end of the trial.

Data on payments made to trial participants, whether triggered by the card reader or manually, were extracted from the trial portal. These data are reported in the process evaluation paper, along with details of the challenges with the electronic system.[27]

### Sample size

We analysed all primary outcomes jointly, to maximise power, aid interpretation and minimise testing (see the Statistical analysis section). However, for our sample size calculation, we considered the power to detect an effect of the intervention on one primary outcome, which we also assumed to be a binary indicator that all attendances were made, as this is simple and conservative in the power achieved. The expected prevalence of these indicators in the control arm ranged between 30% and 80%. In the absence of specific information on the likely intra-cluster correlation (ICC) we considered a range between 0.005 (low) and 0.025 (moderate). Our planned sample size was 48 clusters covering the catchment areas of selected level 2 and 3 health facilities (24 per arm) and an average cluster size of 150 participants. At a low ICC, the design effect (DE) would be 1.745 and hence the effective sample size (ESS) would be 2063 participants per study arm. At a moderate ICC, the DE would be 4.725 and hence the ESS 762 per arm. Power to detect absolute differences is lowest when the prevalence is 50% and highest when the prevalence is either high (towards 100%) or low (towards 0%). Here, we considered the prevalence in the control arm to range between 50% ('worst-case scenario') and 80% ('best-case scenario'). We considered the standard 5% significance level. If the prevalence of the outcome is 50% in the control arm, the sample size provides 80% power to detect an improvement to 54.5% in the intervention arm if the ICC is low and 57.5% if the ICC is moderate. If the prevalence of the outcome is 80% in the control arm, the sample size provides 80% power to detect an improvement to 83.5% if the ICC is low and 85.5% if the ICC is moderate.

### Data definitions and processing

For the purpose of statistical analysis, eligible ANC visits were defined based on the recorded 'next scheduled visit date' and noting that this was typically 4 weeks later each time. For each recorded visit starting with enrolment, we evaluated whether the next observed ANC visit was within ±2 weeks of the 'next scheduled visit date': recording a successful ANC visit if yes, but a missed visit if not. However, we did not count the visit as either successful or missed if the next observed visit was more than 2 weeks early compared with the next scheduled visit or delivery occurred within 2 weeks of a 'missed' scheduled visit. Whether that next visit is early, on time, or late, we assessed subsequent visit attendances in the same way. We created hypothetical scheduled visits every 4 weeks for any gaps in observations, judged according to the same criteria. The 'successful' and 'missed' appointments were then summed over all scheduled and hypothetical appointments for each woman.

For the primary outcomes of PNC visits, we defined a binary indicator of one or more PNC visit 4–12 months post partum. This was used to provide a simple indicator of engagement with PNC for each woman, given that the appropriate number of PNC visits may differ between women. We defined child immunisation visits as the total number of visits recorded post partum (excluding vaccination at delivery, but without other time restrictions) truncated at a maximum of four, since that is the typical number required for full immunisation.

Retention in the full continuum of ANC, perinatal and PNC (a secondary outcome) was defined as a binary indicator of attendance of all eligible ANC, child immunisation and PNC visits and delivery at a healthcare facility for each woman and is available for those women who brought their clinic book for data extraction.

### Statistical analysis

The primary outcomes of attendance of eligible ANC and child immunisation appointments comprise repeat binary observations for each individual woman, while the primary outcomes of delivery at a health facility and PNC attendance between 4 and 12 months are each single binary variables for each woman. A summary OR is presented as the main effect measure for the trial, estimated from a model that assumes the OR is the same across the four primary outcomes. We also report separate effect estimates for each primary outcome. A mixed effects logistic regression model was used to jointly analyse the observed primary outcomes. This approach assumes that, given their ANC attendances (recorded for nearly all women), whether a woman did or didn't bring their clinic book for extraction of the other outcomes was unrelated to the values of those outcomes. At the level of each woman correlated random effects were specified for (1) attendance of ANC clinic visits and (2) grouped outcomes of delivery at a healthcare facility, attendance of vaccination visits and attendance of at least one PNC clinic visit at 4–12 months post partum. Clinic-level random effect terms were defined for (1) ANC visits, (2) delivery at a healthcare facility and (3) PNC and child immunisation visits, with unrestricted correlations between these.

The secondary outcomes of counts of ANC clinic, PNC clinic and child immunisation visits were analysed using a multivariate Poisson mixed effects model, with random intercept terms at patient and the clinic levels. Marginal mean differences in counts between intervention groups were estimated. The secondary outcome of GA at enrolment to the trial was analysed using a linear mixed effects model, with random intercept term at clinic level. Retention in the full continuum of care was analysed using a logistic regression mixed effects model, with random intercept term at clinic level. As the completion rate of the follow-up survey was lower than expected, the secondary outcome data obtained is reported in a descriptive summary but not compared between trial arms.

Our main analyses are conducted as randomised (ie, intention to treat) but a 'per-protocol' style sensitivity analysis of the primary outcomes was also prespecified. As there was not a clear division between clinics that did and did not achieve the intended payment schedule, the sensitivity analysis included the 12 intervention clinics and the 12 control clinics with the highest proportion of enrolled women with payment within 31 days of first ANC clinic visit.

All analyses were adjusted by the baseline maternal parity ('0' vs '≥1'), by the presence of any maternal medical conditions leading to classification of the pregnancy as high-risk (HIV with or without antiretroviral therapy (ART), diabetes, hypertension, malaria, each coded with separate indicator variables) and by the clinic-level variable of subcounty. Adjusted effect measures are considered the primary effect measures though unadjusted effect estimates are also reported. The analysis followed a prespecified SAP, which was finalised after data collection but prior to any unblinded analysis. Selection of maternal characteristics as adjustment variables was based on their inclusion in the core enrolment dataset, the associated absence of missing data for these items and their potential to predict the outcomes of interest. Subcounty was included as an adjustment variable because of its use as a stratification factor in the randomisation process for the study.

As recommended by the MRC framework for the evaluation of complex interventions,[28] a detailed process and economic evaluation will be published in two forthcoming papers.[27 29]

## RESULTS
The trial was conducted in 24 intervention and 24 control clinics and enrolled a total of 2522 women at intervention clinics and 2949 at control clinics over a period from May 2017 to December 2019 (figure 1). Only 11 eligible women declined enrolment at intervention clinics and 58 at control clinics. Based on the background data on ANC attendance in the study region in 2015, it was expected that each of the 48 facilities would recruit 150 participants into the trial, meeting the target sample size of 7200 eligible women during the trial period. However,

enrolment stopped before the target sample size of 7200 could be reached due to delays arising from the nurses' strike during which enrolment was paused at many clinics (see, eg,[30] who discussed the strike and its impacts on healthcare delivery), and as the trial was intended to run until 2018 initially. The vast majority (5388 or 98.5%) of women had data on ANC attendances, but data from maternal clinic books on all primary outcomes were available in a minority of women (2262/5388, 42.0%). Sociodemographic characteristics for all enrolled women are presented in table 1, with cluster-level summaries in online supplemental table S1. Baseline survey data were available in 4313/5471 (78.8%) women, and a summary of selected fields by arm is presented in table 2. Tables 1 and 2 demonstrate very good balance between arms.

## Primary and secondary outcomes
The proportion of eligible ANC appointments attended was significantly higher in the intervention arm compared with control (67% vs 60%; adjusted OR (aOR) 1.90; 95% CI 1.36 to 2.66). A smaller increase was also demonstrated in the proportion of eligible immunisation appointments attended (88% vs 85%; aOR 1.74; 95% CI 1.10 to 2.77). For the other primary outcomes reporting was similar between arms (table 3). The pooled aOR for the intervention effect giving a summary measure across all primary outcomes was 1.64 (95%CI 1.28 to 2.10), p<0.001. The intervention effect on the number of eligible ANC attendances, expressed as an adjusted marginal change (table 3), was an increase of 0.31 (0.15 to 0.47). The adjusted marginal change in eligible immunisation attendances was not significant (0.14, −0.12 to 0.41). Increases in attendances were seen for all visit types when the eligibility requirements defined for the primary analysis were removed, thereby considering all healthcare visits (including any unscheduled visits). The intervention had no effect on the timing of first ANC visit, the mean GA at enrolment was 22.2 weeks for intervention and 22.3 weeks for control, or quite a few weeks after the recommended first visit by the WHO[31] but consistent with other studies in low-income and middle-income countries.[32] Postnatal surveys at 5–18 months after delivery were completed by a minority of women, selected outcomes are reported by arm in table 4. Maternal and perinatal mortality were not systematically recorded, but the available data on these outcomes are summarised in online supplemental appendix S2. The ICC was 0.028 for the primary outcome of ANC visits, 0.012 for delivery at a health facility, 0.087 for attendance of at least one eligible PNC visit and 0.011 for immunisation visits.

## Sensitivity analysis
A sensitivity 'per protocol' analysis of the primary outcomes was prespecified to only include those clinics and periods for which payments were being processed. However, as the correct functioning of the payment systems did not follow clear temporal divisions, the sensitivity analysis was conducted including those intervention

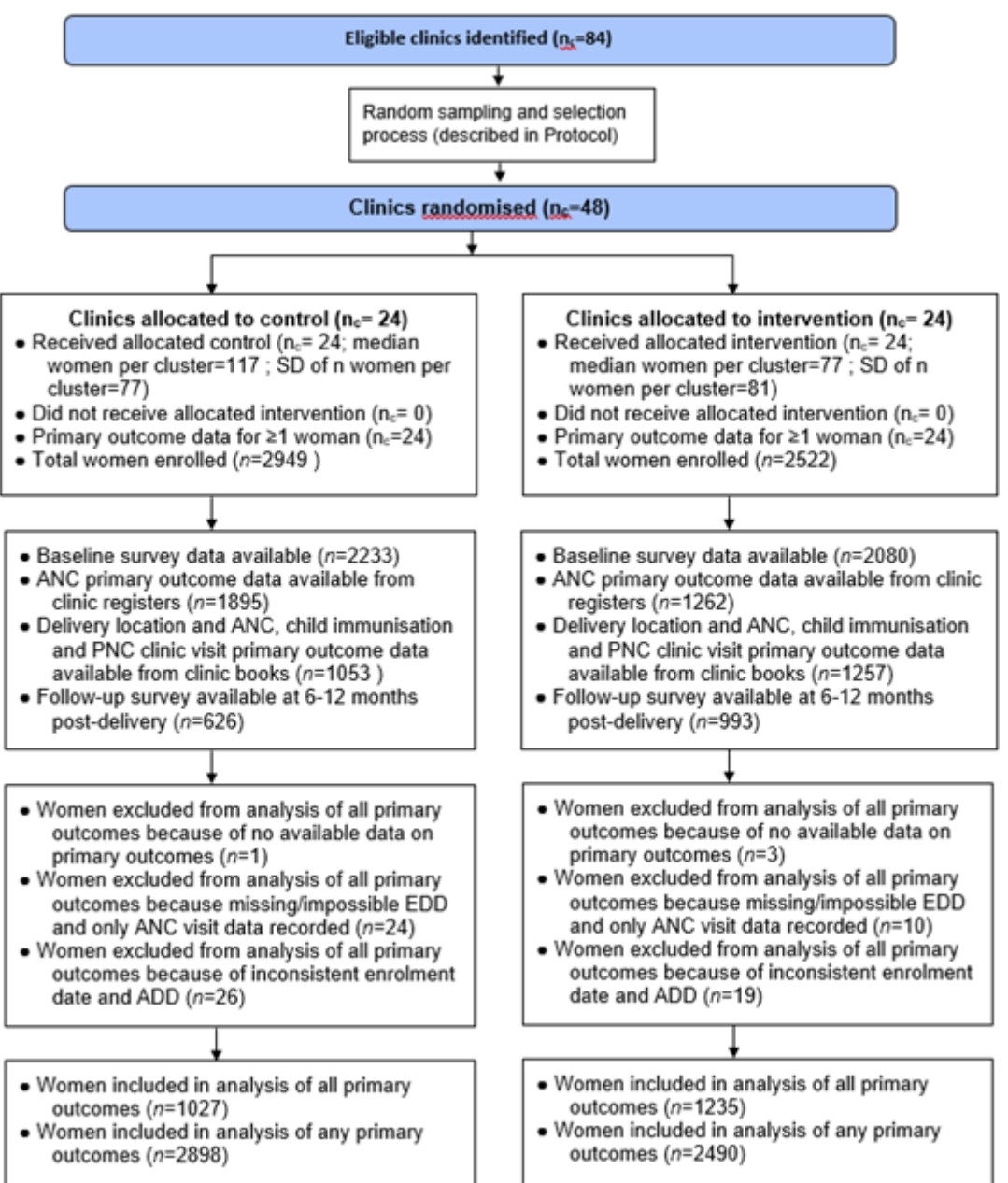

**Figure 1** Flow diagram of enrolment and inclusion in analyses by clinic randomisation status. n values refer to women and $n_c$ to clinics. ADD, actual date of delivery; EDD, expected date of delivery.

and control clinics with a proportion of women with prompt payment (within 31 days) of their first ANC visit above the median within each arm.

We evaluated prompt payment from first visit among the 4156 women with at least one ANC visit included in the primary analysis. There was a prompt payment in 743/2141 (34.7%) women in the control clinics and in 943/2015 (46.8%) in the intervention clinics (online supplemental figure S1). Across control clinics, the median proportion with prompt payment for first visit was 32.3%, with IQR 17.8%–46.1% (online supplemental figure S2). Across intervention clinics, the median proportion with prompt payment was 42.4%, with IQR 27.0%–50.4% (online supplemental figure S3).

In the sensitivity analysis, there was no observed positive effect of the implementation on ANC attendance, and the pooled estimate of the intervention effect was negative but not statistically significant (online supplemental table S2). As this finding was surprising given the main results, we investigated the cluster-level association between prompt payment and ANC attendance. This revealed a stronger correlation in the control clinics than in the intervention clinics (online supplemental figure S4). Consequently, in our sensitivity analysis in terms of ANC attendance rates, we were unexpectedly comparing high performing control clinics with typical intervention clinics, which explains the change to the intervention effect from the main analysis.

**Table 1** Baseline and pregnancy characteristics of the enrolled women included in the primary analysis, obtained from enrolment data, clinic book and clinic registry data

| Variable | | Control clinic, n (%) or median (IQR) | Intervention clinic, n (%) or median (IQR) |
|---|---|---|---|
| Baseline characteristics | | | |
| Age | Years (median (IQR) (n)) | 26 (22–31) (2738) | 26 (22–31) (2349) |
| Parity | 0 | 606 (21) | 488 (20) |
| | 1 | 698 (24) | 591 (24) |
| | 2 | 587 (20) | 485 (19) |
| | ≥3 | 1007 (35) | 926 (37) |
| GA at enrolment | Weeks+days (median (IQR) (n)) | 22+4 (17+4 to 27+2) (2898) | 22+3 (17+2 to 27+0) (2487) |
| HIV status | Negative | 2481 (86) | 2142 (86) |
| | Positive, on treatment | 398 (14) | 322 (13) |
| | Positive, not on treatment | 19 (1) | 26 (1) |
| Diabetes | No | 2889 (99.7) | 2482 (99.7) |
| | Yes | 9 (0.3) | 8 (0.3) |
| Hypertension | No | 2892 (99.8) | 2485 (99.8) |
| | Yes | 6 (0.2) | 5 (0.2) |
| Malaria | No | 2603 (90) | 2164 (87) |
| | Yes | 295 (10) | 326 (13) |
| Total high-risk pregnancies | No | 2245 (77) | 1850 (74) |
| | Yes | 653 (23) | 640 (26) |
| Pregnancy characteristics | | | |
| GA at delivery | Weeks+days (median (IQR) (n)) | 39+3 (37+0 to 41+1) (1004) | 39+0 (36+4 to 41+0) (1206) |

GA, gestational age.

## DISCUSSION

In this paper, we evaluated the impact of a demand side financing intervention, using CCTs, on the retaining pregnant women in the continuum of care from their first ANC visit until their children reach 1 year of age in rural Kenya. Previous evaluations of CCT programmes in the Sub Saharan Africa region have focused on either increasing ANC visits,[33] institutional deliveries[18 33] or PNC visits.[34] Two other studies in the region evaluated the impact of demand side financing on the retaining women in the continuum of care. For the continuum of care, one study evaluated the impact of a subsidised reproductive health voucher programme and the introduction of free maternity services in government facilities on ANC visits, facility birth and PNC visits in Kenya.[35] Another study[36] examined the impact of a national CCT pilot programme on the continuum of care in rural Nigeria. To our knowledge, this is the first evaluation of the impact of CCTs on retaining pregnant women in the continuum of care in Kenya and provides crucial evidence to inform policy and practice related to demand side financing mechanisms for improving maternal and newborn health outcomes.

### Increased ANC clinic attendance and child immunisation appointments

Our main finding suggests that the intervention led to a modest increase in ANC clinic attendance and child immunisation appointments. This is consistent with evidence on the impact of demand side financing programmes on ANC service utilisation from the sub-Saharan African region but less so with the evidence on child immunisation. For example, a study set in Kenya[35] found that a subsidised reproductive health voucher programme and free maternity services improved early initiation of ANC as well as continuous use of care among ANC attendees in government facilities. Others[33 36] also found that CCT programmes increased ANC attendance in rural Kenya and Nigeria, respectively. This implies that demand side financing interventions, whether using CCTs or vouchers can increase ANC service utilisation, though the size of the impact might vary. It is possible that the increase in ANC service utilisation might be greater if financial incentives such as CCTs and vouchers are combined with other policy measures such as free maternity services.

**Table 2** Sociodemographic characteristics of the enrolled women from baseline survey

| Variable | Control clinic, n (%) | Intervention clinic, n (%) |
|---|---|---|
| Enrolled women | | |
| Total n | 2949 | 2522 |
| Baseline survey | | |
| Available | 2233 (76) | 2080 (82) |
| Missing | 716 (24) | 442 (18) |
| Self-rated maternal health | | |
| Very good | 4 (0.2) | 3 (0.1) |
| Good | 771 (35) | 718 (35) |
| Moderate | 1445 (65) | 1351 (65) |
| Bad | 12 (0.5) | 6 (0.3) |
| Very bad | 0 (0) | 0 (0) |
| Missing* | 1 (0.04) | 2 (0.1) |
| Maternal education level | | |
| None or only literacy | 15 (0.7) | 17 (0.8) |
| Primary incomplete | 631 (28) | 540 (26) |
| Primary complete | 860 (39) | 796 (38) |
| Secondary incomplete | 352 (16) | 343 (16) |
| Secondary complete | 292 (13) | 292 (14) |
| University/college | 79 (4) | 91 (4) |
| Don't know/other/missing* | 4 (0.2) | 1 (0.05) |
| Mode of travel to facility for enrolment visit | | |
| Public transport, for example, bus | 727 (33) | 718 (35) |
| Mini bus taxi | 0 (0) | 0 (0) |
| Metered/taxi | 4 (0.2) | 7 (0.3) |
| Walking | 1485 (67) | 1341 (64) |
| Car | 1 (0.04) | 1 (0.05) |
| Other | 16 (0.7) | 11 (0.5) |
| Missing* | 0 (0) | 2 (0.1) |
| Travel time to facility for enrolment visit | | |
| <1 hour | 1444 (65) | 1314 (63) |
| 1–2 hours | 741 (33) | 699 (34) |
| 2–3 hours | 43 (2) | 60 (3) |
| >3 hours | 4 (0.2) | 5 (0.2) |
| Missing* | 1 (0.04) | 2 (0.1) |

*Of those women with baseline survey recorded.

As this was a cluster RCT, it was possible that knowledge of the incentives spread in the community and women could have attend earlier to collect more incentives. However, the results show that women (in both arms) attended their visit in week 22 on average. According to the WHO recommendations, the first ANC visit should be scheduled between week 8 and 12 of the pregnancy.[31] Late attendance for ANC visits was also found in other studies,[32 37 38] and the consequences of that late first visit require further investigation, and further research could be done in how to incentivise early attendance.

### Limited effect on facility delivery and PNC visits

The Nigerian CCT programme did not find any impacts on neonatal immunisation.[36] In Zimbabwe, little improvement was found in immunisation among children under 5 years of age.[34] This is in contrast with our findings. We did not observe a significant impact on facility delivery, which is consistent with the findings from[36] but not with other studies[18 39] that found a positive impact of CCTs on facility delivery. We did not find a clear intervention effect on the proportion of women with at least one PNC visit at 4–12 months, but women did have higher numbers of total PNC visits in the intervention clinics. This suggests further research into unpacking why financial incentives do not have as consistent an impact on facility delivery and PNC visits or child immunisation in comparison to the effect on ANC service utilisation.

### Challenges with the trial

We faced two major and connected challenges related to the technical functioning of the card system and a delay in the transfer of payments, the latter being common for CCT programmes.[40] The touching in of the Afya card reader was intended to record the visit and automatically trigger payments to participants. However, only 26% of payments were triggered automatically (further details in Dickin *et al*[27]). The remaining transfers required involvement of the field implementation partner to manually record a visit and trigger a transfer. This caused several delays in payments being made to participants, often over months. Other challenges such as healthcare staff not tapping the cards to avoid conflicts with participants over delayed payments, reluctance of new staff at facilities to participate due to challenges with delayed transfers; the card reader being locked by the main staff member actively involved in intervention to avoid theft but limiting use by other healthcare staff contributed to the intervention not being implemented as intended. All these factors, linked to the technology and delay in payments could have diluted the impact of the intervention and prevented service utilisation in the latter stages of pregnancy or after babies were born.

Our findings of modest increases in scheduled ANC and immunisation appointment attendance, need to be interpreted in the context of the implementation challenges described, noting that most of these are partly inherent to the intervention we evaluated given the resources available and the trial context. Although implementation of the intervention could have been enhanced, for example, by either more frequent regular visits to facilities to support clinic staff or by incentivising clinic staff more to process payments throughout a woman's care, these models may be unsustainable at scale. We saw evidence of a modest intervention effect on counts of all attendance types other than delivery when lifting the eligibility requirements for the analysis and thereby potentially

**Table 3** Effect of conditional cash transfers on primary and secondary outcome measures

| Primary outcome measures | Control clinic | Intervention clinic | | |
|---|---|---|---|---|
| | n/N (%) | n/N (%) | OR (95% CI) | aOR (95% CI); P value |
| Attendance at eligible ANC clinic appointments (following scheduled visits)* | 5827/9736 (60) | 5741/8595 (67) | 1.95 (1.39 to 2.72) | 1.90 (1.36 to 2.66); p<0.001 |
| Delivery at health facility† | 945/1027 (92) | 1115/1238 (90) | 0.59 (0.26 to 1.35) | 0.58 (0.25 to 1.33); p=0.20 |
| Attendance at one or more eligible PNC clinic appointment (4–12 months after delivery)‡ | 831/1027 (81) | 1016/1235 (82) | 1.26 (0.75 to 2.12) | 1.25 (0.74 to 2.10); p=0.40 |
| Attendance at child immunisation appointments (capped at 4)† | 3498/4108 (85) | 4353/4952 (88) | 1.76 (1.10 to 2.80) | 1.74 (1.10 to 2.77); p=0.02 |
| Pooled intervention estimate | – | – | 1.69 (1.32 to 2.17) | 1.64 (1.28 to 2.10); p<0.001 |
| **Secondary outcome measures** | **Control clinic** | **Intervention clinic** | | |
| | n/N (%) | n/N (%) | | aOR (95% CI) |
| Attendance at all eligible ANC and PNC visits, child immunisation appointments and delivery at a healthcare facility (per woman) | 480/1027 (47) | 632/1235 (51) | | 1.14 (0.82 to 1.57) |
| | **Mean (95% CI)§, median (IQR)** | **Mean (95% CI)§, median (IQR)** | | **Average difference in mean (95% CI)§** |
| **Visit counts (eligible for primary outcome)** | | | | |
| Total attendances at eligible ANC clinic appointments (following scheduled visits)* | 2.04 (1.93 to 2.14) 2 (0–3) | 2.34 (2.22 to 2.47), 2 (1–3) | | 0.31 (0.15 to 0.47) |
| Total attendances at eligible child immunisation appointments (capped at 4)† | 3.33 (3.14 to 3.52), 4 (3–4) | 3.48 (3.29 to 3.67), 4 (4–4) | | 0.14 (−0.12 to 0.41) |
| **Visit counts (no eligibility criteria applied)** | | | | |
| Total attendances at ANC clinic appointments‡ | 2.05 (1.93 to 2.18), 2 (0–3) | 2.42 (2.27 to 2.57), 2 (1–4) | | 0.37 (0.18 to 0.56) |
| Total attendances at PNC clinic appointments† | 4.17 (3.90 to 4.44), 4 (2–7) | 4.76 (4.46 to 5.06), 5 (2–7) | | 0.58 (0.19 to 0.98) |
| Total attendances at child immunisation appointments† | 3.64 (3.40 to 3.88), 4 (3–5) | 4.00 (3.75 to 4.26), 4 (4–5) | | 0.36 (0.02 to 0.71) |
| | **Mean (95% CI)** | **Mean (95% CI)** | | **aΔ (weeks) (95% CI)** |
| GA at enrolment (weeks)* | 22.3 (21.9 to 22.7) | 22.2 (21.8 to 22.6) | | −0.1 (−0.6 to 0.5) |

*Available for all women included in primary analysis except three (with data from clinic book but no available expected date of delivery (EDD) or actual date of delivery (ADD)).
†Available for 2265/5388 women with data obtained from clinic books.
‡Available for 2262 women with data obtained from clinic books and with available ADD or EDD. Available data for attendances and eligible appointments are summed over all women for the n/N values.
§Marginal value derived from multivariate Poisson mixed effects model applied to visit counts, estimated over the baseline characteristics of all 5388 women with data for at least one of the primary outcomes.
¶Available for all women included in primary analysis.
ANC, antenatal care; aOR, adjusted OR; aΔ, adjusted difference in mean; PNC, postnatal care.

including some unscheduled visits and visits additional to the expected maximum required. These findings may reflect in part an increase in visits made without serious health concerns where the cash transfer is a major motivation, though unscheduled visits were not eligible for payments. We conducted a per-protocol sensitivity analysis, but this was ultimately unhelpful since it led to an unfair comparison between arms, and so we are unable to quantify the intervention effects that might be expected had implementation been more complete.

**Table 4**  Data from follow-up surveys completed 5–18 months after delivery

| Variable | | Control clinic | Intervention clinic |
|---|---|---|---|
| Enrolled women | Total n | 2949 | 2522 |
| Survey answers 5–18 months after delivery | | | |
| Survey | Available | 626 (21) | 993 (39) |
| | Missing | 2323 (79) | 1529 (61) |
| Time from delivery at survey (days) | | 379 (273–463) | 375 (273–469) |
| Self-rated maternal health | Very good | 70 (11) | 185 (19) |
| | Good | 144 (23) | 190 (19) |
| | Moderate | 37 (6) | 42 (4) |
| | Bad | 4 (0.6) | 3 (0.3) |
| | Very Bad | 0 (0) | 0 (0) |
| | Not asked in survey included | 359 (57) | 548 (55) |
| | Missing* | 12 (2) | 25 (3) |
| Exclusive breast feeding to 6 months | Yes | 244 (39) | 396 (40) |
| | No | 11 (2) | 24 (2) |
| | Not asked in survey included | 359 (57) | 548 (55) |
| | Missing* | 12 (2) | 25 (3) |
| Family planning advice at last clinic visit | Yes | 539 (86) | 864 (87) |
| | No | 58 (9) | 78 (8) |
| | Missing* | 29 (5) | 51 (5) |
| Current method to prevent pregnancy | None | 94 (15) | 137 (14) |
| | NA: pregnant | 3 (0.5) | 4 (0.4) |
| | Yes: contraceptive | 433 (69) | 723 (73) |
| | Yes: natural methods | 57 (9) | 49 (5) |
| | Yes: other | 9 (1) | 29 (3) |
| | Missing* | 30 (5) | 51 (5) |

Data shown as n, n (%) or median (IQR). Data were available for some women for both the planned '6-month survey' and the planned '12-month survey' (although the actual timing was not necessarily as planned), and in these the latter was used if completed 5–18 months after delivery as it contained questions on self-rated maternal health and age at end of exclusive breast feeding.
*Of those women with relevant survey data available.

Our randomisation process was based on selecting 48 facilities to participate from a shortlist of 60 and simultaneously randomising these to intervention or control. Where the catchment area for a selected facility was found to overlap with a previously selected facility it was replaced. Although we believe the randomisation process was implemented objectively, we acknowledge that there could have been some subjectivity in deciding whether catchment areas overlapped and since the allocation to intervention or control was already revealed at this point it is theoretically possible that bias was introduced.

The collection of outcome data as originally planned was unfeasible and, while it is a real strength that we managed to collect ANC attendance data for almost all trial participants, it is a limitation that we managed to collect data on the other primary outcomes for only a minority of women and less commonly in the control arm, and no data on attendance for referrals. The potential bias is however limited to a degree by our approach to analysis in which all outcomes are modelled simultaneously. Although the telephone follow-up surveys proved challenging these data are not central to the analyses and interpretation presented here. We did not recruit to our original sample size target but obtained outcomes from all clusters.

## CONCLUSIONS

This trial has demonstrated modest benefits from a CCT intervention, that was affected by technical and other implementation challenges. Further research is needed to address how to design a more robust process for registering attendances and ensuring rapid payment of CCTs to ensure women have confidence in receipt of CCTs for future attendances. This could impact incentivise women to attend visits earlier in their pregnancy as well.

**Acknowledgements** The authors thank the following people for their contribution to the trial: Caroline Ochieng, Alie Eleveld, Geordan Shannon, Stacey Noel, Matthew Fielding, Sangoro Onyango and Sarah Odera.

**Contributors** FV was the Principal Investigator, leading study implementation from May 2018 to June 2020 and acted as guarantor. OS, TP, and AC were the trial statisticians, with AC leading the design of the data analysis methods and interpretation of the research findings, and OS undertaking the analysis. AO was the Trial Coordinator, leading field implementation under supervision of AM. SD, JS-W, NB, HH-B, TP led specific components of the trial such as the process evaluation and the economic evaluation and contributed to the overall research methods and design. The entire team contributed to the interpretation of the research findings and the writing of the paper. All authors read and approved the final manuscript. FV acts as a gaurantor.

**Funding** The authors thank the Bill and Melinda Gates Foundation for the funding for this project (grant number: OPP 1142564) and the Swedish International Development Cooperation Agency for providing cofinancing (grant number: not applicable). The funders were not involved in the design, implementation, data collection, analysis, writing and decision to submit the paper for publication.

**Competing interests** AC who is associate editor of Sexually Transmitted Infections.

**Patient consent for publication** Consent obtained directly from patient(s)

**Ethics approval** This study involves human participants and was approved by MSU/DRP/MUERC/00294/16; Maseno University, Kenya Participants gave informed consent to participate in the study before taking part.

**Provenance and peer review** Not commissioned; externally peer reviewed.

**Data availability statement** Data are available upon reasonable request.

**ORCID iDs**
Fedra Vanhuyse http://orcid.org/0000-0002-9283-1914
Neha Batura http://orcid.org/0000-0002-8175-8125
Hassan Haghparast-Bidgoli http://orcid.org/0000-0001-6365-2944
Andrew Copas http://orcid.org/0000-0001-8968-5963

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
