## [Reviewer comments · BMJ Open]

ARTICLE DETAILS

TITLE (PROVISIONAL)	Effectiveness of conditional cash transfers (Afya credits incentive) to retain women in the continuum of care during pregnancy, birth and the postnatal period in Kenya: a cluster randomised trial
AUTHORS	Vanhuyse, Fedra; Stirrup, Oliver; Odhiambo, Aloyce; Palmer, Tom; Dickin, Sarah; Skordis-Worrall, Jolene; Batura, Neha; Haghparast-Bidgoli, Hassan; Mwaki, Alex; Copas, Andrew

VERSION 1 – REVIEW

REVIEWER	Yu Gao Charles Darwin University
REVIEW RETURNED	29-Aug-2021

GENERAL COMMENTS	The authors have clearly followed the STROBE and addressed every aspect of the guidelines with details. The study was well-designed and the manuscript is a very important contribution to the literature of demand side financing mechanisms for improving maternal and child health in low-income countries. When there are differences between protocol and their data collection and analysis in the manuscript, the authors have explained every detail. I don't have any other comments to improve the paper and I would like to recommend to accept the manuscript for publication.
--

REVIEWER	Margaret McConnell Harvard University T H Chan School of Public Health, Global Health and Population
REVIEW RETURNED	14-Sep-2021

GENERAL COMMENTS	This paper reports the results of a randomized experiment of the impact of conditional cash transfers for attendance at maternal care appointments including ANC, delivery, PNC and immunization. The analysis finds that CCTs increased rates of antenatal care and child immunization but finds limited impacts on other outcomes, potentially due to challenges in implementation. I commend the authors for transparency surrounding the significant implementation challenges that occurred during this study. Unfortunately, many of the implementation challenges pose challenges in analyzing the results which I discuss including some suggestions for how to focus analysis on areas where challenges of missing data are less severe. The paper should report a clear consort diagram. Within the context of this experiment, missing outcome data should be reported transparently for different outcomes and across treatment groups as there are significant concerns about potential differential attrition given the difference in incentives for documentation of visits by the
--

	treatment group compared to the control group. Given the challenges of data collection and abstraction, I would think that focusing primarily on the most complete outcome (ANC visits) would be advisable. The strategy of “imputing” missing outcomes seems to really lose the value of a randomized controlled trial and imposes many extremely strong assumptions. I would downplay in the analysis and discussion what can be learned from any outcome where a significant share of the observations cannot be observed. Much can still be learned from the ANC visits outcome which is the only complete outcome. I also have a few specific comments and questions:  - I have trouble understanding how the data collection was supposed to work. It seems there would have been no card readers in control facilities so this data source would not have been suitable for measuring outcomes. How was outcome data on visitation supposed to be collected? - More details are needed to understand the site selection and randomization. For example did the replacement draws occur at the stage of site selection or was a replacement draw allowed during the randomization process? The latter would indicate some concern about how much discretion there is surrounding the definition of overlapping catchment areas. Were the catchment areas defined in advance of this public meeting? - More discussion is needed of why the trial should have lead to an improvement in the timing of ANC initiation. Wouldn't mothers have been sensitized to the CCTs at their first visit? Why should the intervention affect the timing of the first visit?
--	--

REVIEWER	Avishek Hazra Population Council New Delhi
REVIEW RETURNED	22-Sep-2021

GENERAL COMMENTS	Abstract  • “An electronic system was to capture the visits and trigger automatic payments to the participant” – The authors may explain what is meant by automatic payments. Were payments made to participant’s bank accounts! • Under the description of ‘interventions’, three suggestions: 1) The authors may like to present the incentive amount in USD, for international reader’s ease, 2) apart from attendance, was there any other condition for receipt of payment (for example, household economic condition or parity or anything else)! If so, please mention, and 3) mention how much amount a woman can receive maximum (or mention how many visits a woman can attend in the continuum of care) in the intervention and control area. • “Primary outcomes were ANC attendances, delivery at facility, any PNC 4-12 months after delivery, and childhood immunization”. The authors need to clarify if a woman received PNC within 4 months after delivery, whether/why she was not treated as a positive outcome. Also, please specify what does ‘childhood immunization’ mean – is it age-appropriate immunization, any immunization, or complete immunization and also this indicator is calculated for children of which age? • “Secondary outcomes include total number of visits attended” – please clarify and add in the abstract if total number of visits refers to all visits to the health facility for ANC visit to child immunization. Main text
--

	 • Page 5: lines 10-14 – there is duplication of same sentence. Revise this. • The randomization process needs to be articulated more clearly. In line 15, it says ‘implementing partner wrote the names of 60 shortlisted facilities’, while line 9 says ‘units of randomization were 48’. “...included them in transparent boxes, one for each sub-county” – no mention of how many sub-county, which I guess is a strata. Why stratification was needed! • There is no mention about the sample size calculation or justification of a sample of 48 clusters. Even if reference to the protocol is given, it merits to write a few lines about these.  • “...before each facility was formally recruited, a check was made that the catchment area for the facility did not overlap with those on any facilities already recruited” – this is good, but in reality, women from one catchment area (say control area) may avail health services from a health facility which is nearer to their homes, but falls in other catchment area (intervention area). How did the study control that contamination! • “Health facility staff determined whether a pregnant woman met the study eligibility criteria by administering screening questions...” – it may be worth mentioning here the study eligibility criteria and mention the screening questions in appendix. • “Individual consent was required for trial participation, and refusals were recorded” – please specify if consent were in local language and if it was written or verbal/oral consent. • Use of English is mixed – British and US English (for example, immunization and immunisation) both have been used. The authors may follow one style throughout the paper, as per BMJ guideline • Page 7, lines 8-9: “Our planned sample size was 48 clusters (24 per arm) and an average cluster size of 150.” Clearly state what is a ‘cluster’ in this case and what does the cluster size refer here! • Page 7, lines 57-60 – “A mixed effects logistic regression model was used to jointly analyse the primary outcomes, which implicitly imputes the missing information on facility delivery, PNC and child immunisation visits based on ANC attendance for women who did not bring their maternal clinic book for data extraction” – a clearer description on how missing data were handled and associated imputation process is required. • Page 8, lines 3-8 – the sentences may be refined to articulate more clearly in a lucid language about the random effects at individual and clinic level.  • “The secondary outcomes of counts of ANC clinic, PNC clinic and child immunisation visits were analysed using a multivariate Poisson mixed effects models” – since these events (visits) are not rare events, what was the rationale of using Poisson model! • “All analyses were adjusted by the baseline maternal parity (‘0’ vs ‘≥1’), by the presence of any maternal medical conditions leading to classification of the pregnancy as high-risk (HIV with or without ART, diabetes, hypertension, malaria, each coded with separate indicator variables) and by the clinic-level randomisation stratification variable of sub-county.” – it is important to mention what criteria were followed for selection of variables for adjustment.
--	--

VERSION 1 – AUTHOR RESPONSE

No	Comment	Response
1	Please check for consistency in the primary and secondary outcomes defined in the manuscript, protocols (both the published	Agreed – see also comment 5, 11, 19 and 27

	version and the full protocol) and trial registration page. If you spot any discrepancies, please revise the manuscript and/or update the registry page as appropriate to clarify.	We have acknowledged the one primary outcome for which we collected no data in the abstract and amended text throughout the article concerning 'omitted' outcomes so we feel there should no longer be any inconsistency with the registration. We have highlighted changes throughout the manuscript to correspond to this issue.
2	Please update article title to better indicate the study design – specifically, please specify the continuum of care “during pregnancy, birth and the postnatal period” and please indicate the study design (cluster randomised trial), as per journal style. Eg, “Effectiveness of conditional cash transfers (Afya credits incentive) to retain women in the continuum of care during pregnancy, birth and the postnatal period in Kenya: a cluster randomised trial”	Agreed – changed to: Effectiveness of conditional cash transfers (Afya credits incentive) to retain women in the continuum of care during pregnancy, birth and the postnatal period in Kenya: a cluster randomised trial
3	Please update the “Objectives” section of the abstract to add a very brief sentence of background, before revising the existing sentence to start “We aimed to...”.	Agreed – Given word count limitations, we changed the abstract to: Given high maternal and child mortality rates, we assessed the impact of Conditional Cash Transfers (CCTs) to retain women in the continuum of care (antenatal care (ANC), delivery at facility, postnatal care (PNC) and child immunization).
4	Please update the abstract format to include a “Design” section before the “Setting” section (separating out the design and setting information in the text), as per journal style.	Agreed – Section added with: Design We conducted an unblinded 1:1 cluster-randomized controlled trial. Setting 48 health facilities in Siaya County, Kenya were randomized.
5	To ensure balanced reporting, please ensure that the abstract Results section includes numerical data for all primary outcomes. Secondary outcomes should only be included in the abstract if findings for *all* secondary outcomes can be included in the Results, to ensure results are not being selectively highlighted. If secondary outcomes are not reported in the Results section of the abstract, they should also be	As requested, numerical results for all primary outcomes are now given in the Abstract: “No intervention effect was seen considering delivery at the facility (90 vs 92%; aOR 0.58; 95% CI 0.25-1.33) and any PNC attendance (82 vs 81%; aOR 1.25; 95% CI 0.74-2.10) separately.” The secondary outcomes have been removed from

	removed from the “Primary and secondary outcome measures” section (which should be renamed as “Primary outcome measures”).	the Abstract, as there is not space to list all relevant results. see also comment 1, 11, 19 and 27
6	Please change the heading “Contributor statements” to “Acknowledgments”.	Agreed – changed to: Acknowledgements
7	BMJ Open follows the new ICMJE data sharing policy , which states that as of 1st July 2018 manuscripts submitted to ICMJE journals that report the results of clinical trials must contain a data availability statement that includes answers to the following questions: - Which data in particular will be shared? - Are individual deidentified participant data (including data dictionaries) available? - Are additional, related documents available (e.g., study protocol, statistical analysis plan, etc.)? - When will the data become available and for how long (if relevant)? - By what access criteria will the data be shared (including with whom, for what types of analyses, and by what mechanism)? Please update your ‘Data availability statement’ as needed to make sure that it contains all required information.	Agreed – Changed to: De-identified data may be made available upon request to researchers who provide a scientifically and methodologically sound research proposal and obtain ethical approval for their planned analysis. Proposals should be submitted to the corresponding author. Data that can be shared includes: number of visits made, type of visit, arm and date of the visit. Data dictionaries can be made available, as well as the study protocol and the statistical analysis plan.
8	Where you state “Individual consent was required for trial participation, and refusals were recorded”, please amend to indicate if this was informed consent and to specify the format (eg, written informed consent).	Agreed – see also comment 33 changed to: “oral informed consent was asked in the local language, and then written down on the participant’s enrolment form. Refusals were recorded.”
9	Please use the CONSORT checklist extension for cluster RCTs (a copy should be uploaded indicating the page numbers where the required information is reported).	Agreed – File was already uploaded when the manuscript was originally uploaded. Do let us know if it did not come

		through.
10	The abstract needs to be revised to make it clear that you were not able to collect outcome data as originally planned. So rather than say “despite challenges...” it may be more helpful to state that there were challenges, and to clarify the impact on implementation.	Agreed – Given word count limitations, we changed part of the abstract to: Interventions An electronic system recorded attendance and triggered payments to the participant’s mobile for the intervention arm (4.5 USD), and phone credit for the control arm (0.5 USD). Eligibility criteria were resident in the catchment area and access to a mobile phone. Primary outcomes Primary outcomes were any ANC, delivery, any PNC between 4- 12 months after delivery, childhood immunization, and referral attendance to other facilities for ANC or PNC. Given problems with the electronic system, primary outcomes were obtained from maternal clinic books if participants brought them to data extraction meetings (1257 (50%) of intervention and 1053 (36%) control arm participants).
11	Similarly, please further revise the abstract to be more transparent about the planned outcomes and findings. The secondary outcomes as listed on the trial reg site are missing (maternal deaths, live births, timeliness of visits, self rated wellness, breastfeeding, contraception use and infection screening). These are also not reported in the main body of the paper.	Agreed – See also comment 1,5, 19 and 27 We have removed mention of the secondary outcomes from the Abstract, as there is not space for full details and results. We have now however explicitly mentioned the primary outcome ‘attendance at referrals’ which is not reported because very limited data were collected. We have also added further details regarding deviations from the original study protocol from the Appendix into the main text of the paper under ‘Trial outcomes’ in the Methods: “The primary outcome of attendance at referrals to other facilities for ANC, PNC or child immunization is not reported due to very limited data. The following planned secondary outcomes are reported without formal statistical analysis because of low levels of data completeness: maternal and neonatal mortality, self-rated wellness, exclusive

		breastfeeding and contraceptive use. The following secondary outcomes were dropped because of lack of available data: timeliness of health visits (recorded visits could not consistently be matched up to scheduled dates), and infection screening. These changes were specified in the SAP.” Available data on self-rated maternal health, breastfeeding and contraceptive use are given in Table 4. The available mortality data are provided in Appendix S2 for completeness. We feel that it is appropriate to omit these data from the main text, as the level of ascertainment is unknown.
12	Please explain in the abstract that there were difficulties with the data collection (and eg postnatal surveys were completed by only 1,619/5,471 women (30%)).	Agreed – We have now reported the methods of collecting the primary outcome data in the Abstract together with the percentage of women bring their maternal clinic book for data extraction by arm. Change made to: Interventions An electronic system recorded attendance and triggered payments to the participant’s mobile for the intervention arm (4.5 USD), and phone credit for the control arm (0.5 USD). Eligibility criteria were resident in the catchment area and access to a mobile phone. Primary outcomes Primary outcomes were any ANC, delivery, any PNC between 4- 12 months after delivery, childhood immunization, and referral attendance to other facilities for ANC or PNC. Given problems with the electronic system, primary outcomes were obtained from maternal clinic books if participants brought them to data extraction meetings (1257 (50%) of intervention and 1053 (36%) control arm participants).
13	In the main body of the paper can they explain the sample size: target sample size was 7,200	Based on the background data on ANC attendance in the study region, it was expected that each facility would recruit 150 participants to the trial, giving a total sample of 7,200 eligible women over a period of 24 months. We included a sample size calculation. In a pilot project carried out before this RCT, 200

		participants were recruited from one facility.
14	Were there any ethical issues about paying women in one area and not the other? How did you manage contamination between areas?	The health facilities were selected so there could not be any contamination between the intervention and control arm, as distances between the facilities participating in the trial were too large. When the health facilities were selected (during a public forum with the county government), for each selected facility, the records officer for the subcounty was asked to map the location and catchment area on a large map of the county. If a subsequently selected facility was found to have an overlapping catchment area with a facility previously selected, then the newly selected facility was rejected, and another drawn from the box to take its place. This process continued until 48 facilities had been selected and allocated for the trial. In total, Siaya County had 174 health facilities in 2015 (123 public, 7 non-governmental, 16 faith-based and 28 private)
15	Please consider using subheadings in the Discussion to guide the reader.	Agreed – Added subheadings to indicate findings: Increased ANC clinic attendance and child immunization appointments Limited effect on facility delivery and PNC visits Challenges with the trial
16	The authors have clearly followed the reporting guidance and addressed every aspect of the guidelines with details. The study was well-designed and the manuscript is a very important contribution to the literature of demand side financing mechanisms for improving maternal and child health in low-income countries. When	No change required

	there are differences between protocol and their data collection and analysis in the manuscript, the authors have explained every detail. I don't have any other comments to improve the paper and I would like to recommend to accept the manuscript for publication.	
17	This paper reports the results of a randomized experiment of the impact of conditional cash transfers for attendance at maternal care appointments including ANC, delivery, PNC and immunization. The analysis finds that CCTs increased rates of antenatal care and child immunization but finds limited impacts on other outcomes, potentially due to challenges in implementation.	No change required
18	I commend the authors for transparency surrounding the significant implementation challenges that occurred during this study. Unfortunately, many of the implementation challenges pose challenges in analyzing the results which I discuss including some suggestions for how to focus analysis on areas where challenges of missing data are less severe.	No change required
19	The paper should report a clear consort diagram. Within the context of this experiment, missing outcome data should be reported transparently for different outcomes and across treatment groups as there are significant concerns about potential differential attrition given the difference in incentives for documentation of visits by the treatment group compared to the control group.	Agreed – see also comment 1, 5,11 and 27 The CONSORT checklist was included in the supplementary files with the original files, and Figure 1 is a flow diagram of enrolment and inclusion in analyses by clinic randomization status. This was submitted as well (but not within the manuscript).
20	Given the challenges of data collection and abstraction, I would think that focusing primarily on the most complete outcome (ANC visits) would be advisable. The strategy of “imputing” missing outcomes seems to really lose the value of a randomized controlled trial and imposes many extremely strong assumptions. I would downplay in the analysis and discussion what can be learned from any outcome where a significant share of the observations cannot be observed. Much can still be learned from the ANC visits outcome which is the only complete outcome.	We have retained the results of the analyses as specified in the statistical analysis plan – which was written after data collection but prior to unblinding of the results. However as suggested we now give more emphasis to the findings concerning ANC visits and emphasize that this outcome had near complete data in contrast to the others. In the abstract we now report more clearly the completeness of data for the primary outcomes (see comment 12). We should note that no actual ‘imputation’

		procedures were conducted, but we were aiming to make the point that a joint model for outcomes is equivalent to running an imputation model when data are only missing for outcome variables. To avoid any confusion, we have deleted ‘in which we effectively use the ANC attendance data to predict (i.e., ‘impute’) the other outcomes’ in the Discussion and rephrased the corresponding sentence in the Methods: “A mixed effects logistic regression model was used to jointly analyse the observed primary outcomes. This approach assumes that, given their ANC attendances (recorded for nearly all women), whether a woman did or didn’t bring their clinic book for extraction of the other outcomes was unrelated to the values of those outcomes.”. See also comment 36
21	I have trouble understanding how the data collection was supposed to work. It seems there would have been no card readers in control facilities so this data source would not have been suitable for measuring outcomes. How was outcome data on visitation supposed to be collected?	There were card readers in all facilities, both control arm and intervention arm, and the system worked in the same way: tapping a card on the reader triggered a payment. In the intervention arm, the payment was 450 KSH, in the control arm 50 KSH phone credit. Section revised to: In both trial arms, women were issued with a trial card at recruitment and at all facilities there was a card reader, which provided the connection between the trial card and an online portal which stored participants’ data on visits and payments. Payments to the women were triggered by tapping the card on a card reader, which also logged the visit in an online portal[26]. In the event of problems with the card reader, or if the woman did not bring her card to the appointment, payments could alternatively be processed manually by contacting the implementing partner: once the visit was verified with the facility, the implementing partner entered the visit data in the portal, which would then trigger a payment as well.

22	More details are needed to understand the site selection and randomization. For example did the replacement draws occur at the stage of site selection or was a replacement draw allowed during the randomization process? The latter would indicate some concern about how much discretion there is surrounding the definition of overlapping catchment areas. Were the catchment areas defined in advance of this public meeting?	See also comment 29, 30, 31 and 35 We added some more detail on the site selection and randomization in the section methods: The units of randomisation were Level 2 or 3 health facilities (Dispensaries and Health Centres, respectively). The randomisation of centres was stratified by sub-County and ensured equal allocation to study arms within each stratum without any overlap of catchment areas, as described in detail in the trial protocol[25]. In summary, at a public forum with the county government early 2016, the implementing partner wrote the names of 60 shortlisted facilities on pieces of paper and folded them to hide the names, then included them in transparent boxes, one for each sub-county. Each subcounty had an (even) number of facilities to recruit to the trial proportional to subcounty size. The health management teams from each subcounty selected the pieces of paper, one by one. The first was allocated to intervention, second to control. For each selected facility, county officials from the selected subcounty mapped the location and catchment area of the facility on a large map of the county. If a subsequently selected facility had an overlapping catchment area with a previously selected facility, the newly selected facility was rejected, and another drawn to take its place. This process continued until 48 facilities were selected and allocated for the trial. In the discussion section we now acknowledge that because the determination of whether catchment areas or not was performed after revealing the allocation for the cluster that this could conceivably have introduced bias. We added: “ Our randomisation process was based on selecting 48 facilities to participate from a shortlist of 60 and simultaneously randomising these to intervention or
----	--	--

		control. Where the catchment area for a selected facility was found to overlap with a previously selected facility it was replaced. Although we believe the randomisation process was implemented objectively, we acknowledge that there could have been some subjectivity in deciding whether catchment areas overlapped and since the allocation to intervention or control was already revealed at this point it is theoretically possible that bias was introduced.”
23	More discussion is needed of why the trial should have lead to an improvement in the timing of ANC initiation. Wouldn't mothers have been sensitized to the CCTs at their first visit? Why should the intervention affect the timing of the first visit?	We added a clarifying statement ahead of the discussion of these results: “As this was a cluster RCT, it was possible that knowledge of the incentives spread in the community and women could have attend earlier to collect more incentives. However, the results show that women (in both arms) attended their visit in week 22 on average. According to the WHO recommendations, the first ANC visit should be scheduled between week 8 and 12 of the pregnancy[31]. Late attendance for ANC visits was also found in other studies[32, 37, 38], and the consequences of that late first visit require further investigation, and further research could be done in how to incentivise early attendance.
24	Abstract: “An electronic system was to capture the visits and trigger automatic payments to the participant” – The authors may explain what is meant by automatic payments. Were payments made to participant's bank accounts!	Agreed – Given word count limitations, the abstract was changed to: Interventions An electronic system recorded attendance and triggered payments to the participant's mobile for the intervention arm (4.5 USD), and phone credit for the control arm (0.5 USD). Eligibility criteria were resident in the catchment area and access to a mobile phone.
25	Abstract: Under the description of 'interventions', three suggestions: 1) The authors may like to present the incentive amount in USD, for international reader's ease, 2) apart from attendance, was there any other condition for receipt of payment (for example, household economic condition or parity or anything else)! If so,	Agreed – Given word count limitations, the abstract was changed to:

	please mention, and 3) mention how much amount a woman can receive maximum (or mention how many visits a woman can attend in the continuum of care) in the intervention and control area.	Interventions An electronic system recorded attendance and triggered payments to the participant’s mobile for the intervention arm (4.5 USD), and phone credit for the control arm (0.5 USD). Eligibility criteria were resident in the catchment area and access to a mobile phone. Primary outcomes Primary outcomes were any ANC, delivery, any PNC between 4- 12 months after delivery, childhood immunization, and referral attendance to other facilities for ANC or PNC. Given problems with the electronic system, primary outcomes were obtained from maternal clinic books if participants brought them to data extraction meetings (1257 (50%) of intervention and 1053 (36%) control arm participants).
26	Abstract: “Primary outcomes were ANC attendances, delivery at facility, any PNC 4-12 months after delivery, and childhood immunization”. The authors need to clarify if a woman received PNC within 4 months after delivery, whether/why she was not treated as a positive outcome. Also, please specify what does ‘childhood immunization’ mean – is it age-appropriate immunization, any immunization, or complete immunization and also this indicator is calculated for children of which age?	This has been reworded in the Abstract to ‘any PNC between 4 and 12 months after delivery’ to clarify. The following clarifications have been added to the first paragraph of the ‘Trial outcomes’ subsection of the Methods: “The PNC outcome described in the protocol was the ‘proportion of required postnatal visits honoured after recruitment into the study’. However we have used a simplified outcome here because the required appointment schedule was not recorded for each patient. We also restricted attendances to the period 4-12 months after delivery because on blinded review of the available data, prior to writing the statistical analysis plan (SAP), very few visits prior to 4 months post-delivery were coded as PNC. The vast majority of visits prior to 4 months were recorded as vaccination appointments though it is likely some women also received PNC. Vaccinations over the 12 months after delivery were recorded. The details of vaccinations given were not collected, and so the vaccination outcome is based purely on the number of recorded visits from an expected number of four.” See also comment 36
27	Abstract: “Secondary outcomes include total number of visits attended” – please clarify and add in the abstract if total number of visits refers to all visits to the health facility for ANC visit to child immunization.	Following the request of the Editor (comment 5), we have removed mention of the secondary outcomes from the Abstract, as there is not space for full details and results as given in the main text.

		See also comment 1, 5, 11 and 19
28	Page 5: lines 10-14 – there is duplication of same sentence. Revise this.	Agreed – Duplicate sentence removed
29	The randomization process needs to be articulated more clearly. In line 15, it says ‘implementing partner wrote the names of 60 shortlisted facilities’, while line 9 says ‘units of randomization were 48’. “...included them in transparent boxes, one for each sub-county” – no mention of how many sub-county, which I guess is a strata. Why stratification was needed!	Agreed – See also comment 22, 30, 31 and 35 There were 60 shortlisted facilities from amongst which 48 were finally selected to participate in the trial. Stratification by sub-county was chosen because of possible variation between sub-counties in the outcomes. There 6 sub-counties. Change made to: We conducted a cluster randomized controlled trial, with equal allocation to intervention and control arms, in Siaya county, Kenya. The units of randomisation were Level 2 or 3 health facilities (Dispensaries and Health Centres, respectively). The randomisation of centres was stratified by the six sub-Counties and ensured equal allocation to study arms within each stratum without any overlap of catchment areas, as , as described in detail in the trial protocol[25]. In summary, at a public forum with the county government in 2016 the implementing partner wrote the names of 60 shortlisted facilities on pieces of paper and folded them to hide the names, then included them in transparent boxes, one for each sub-county. Each subcounty had an (even) number of facilities to recruit to the trial proportional to subcounty size. The health management teams from each subcounty selected the pieces of paper, one by one. The first was allocated to intervention, second to control., For each selected facility, county officials from the selected subcounty mapped the location

		and catchment area of the facility on a large map of the county. If a subsequently selected facility had an overlapping catchment area with a previously selected facility, the newly selected facility was rejected, and another drawn to take its place. This process continued until 48 facilities were selected and allocated for the trial.
30	There is no mention about the sample size calculation or justification of a sample of 48 clusters. Even if reference to the protocol is given, it merits to write a few lines about these.	Agreed – See also comment 13 We included the sample size calculation in the Methods. [suggest we delete the etxt below here] Changes made in the results section to include: The trial was conducted in 24 intervention and 24 control clinics and enrolled a total of 2522 women at intervention clinics and 2949 at control clinics over a period from May 2017 to December 2019 (Figure 1). Only 11 eligible women declined enrolment at intervention clinics and 58 at control clinics. Based on the background data on ANC attendance in the study region in 2015, it was expected that each of the 48 facilities would recruit 150 participants into the trial, giving a total sample of 7,200 eligible women during the trial period. However, enrolment stopped before the target sample size of 7,200 could be reached due to delays arising from the nurses’ strike during which enrolment was paused at many clinics (see, e.g.,[30] who discussed the strike and its impacts on health care delivery), and as the trial was intended to run until 2018 initially.
31	“...before each facility was formally recruited, a check was made that the catchment area for the facility did not overlap with those on any facilities already recruited” – this is good, but in reality, women from one catchment area (say control area) may avail health services from	Added a clarification to eligibility: Criteria for enrolment were: women attending their first ANC visit; long-term resident of the catchment area served by the health facility (living in the area for at least 6 months); access to a mobile phone that

	a health facility which is nearer to their homes, but falls in other catchment area (intervention area). How did the study control that contamination!	belongs either to themselves or to a member of their household or person whom they trust. The criterium on residence provided additional assurance that women went to the facility within their catchment area, thereby reducing contamination with other facilities.
32	“Health facility staff determined whether a pregnant woman met the study eligibility criteria by administering screening questions...” – it may be worth mentioning here the study eligibility criteria and mention the screening questions in appendix.	Agreed – Added a reference to the trial protocol and its additional files which contains the participant’s information sheet Changed to: Health facility staff determined whether a pregnant woman met the study eligibility criteria by administering screening questions at the end of her first ANC visit, with the screening questions provided in the trial protocol [25].
33	“Individual consent was required for trial participation, and refusals were recorded” – please specify if consent were in local language and if it was written or verbal/oral consent.	Agreed – see also comment 8 changed to: “oral informed consent was asked in the local language, and then written down on the participant’s enrolment form”
34	Use of English is mixed – British and US English (for example, immunization and immunisation) both have been used. The authors may follow one style throughout the paper, as per BMJ guideline	Agreed – reviewed for consistency (British English)
35	Page 7, lines 8-9: “Our planned sample size was 48 clusters (24 per arm) and an average cluster size of 150.” Clearly state what is a ‘cluster’ in this case and what does the cluster size refer here!	Agreed – See also comment 22, 29, 30 and 31 changed to: Our planned sample size was 48 clusters covering the catchment areas of selected level 2 and 3 health

		facilities (24 per arm) and an average cluster size of 150 participants.
36	Page 7, lines 57-60 – “A mixed effects logistic regression model was used to jointly analyse the primary outcomes, which implicitly imputes the missing information on facility delivery, PNC and child immunisation visits based on ANC attendance for women who did not bring their maternal clinic book for data extraction” – a clearer description on how missing data were handled and associated imputation process is required.	No actual ‘imputation’ procedures were conducted, but we were aiming to make the point that a joint model for outcomes is equivalent to running an imputation model when data are only missing for outcome variables. To avoid any confusion, we have deleted ‘in which we effectively use the ANC attendance data to predict (i.e., ‘impute’) the other outcomes’ in the Discussion and rephrased the corresponding sentence in the Methods: “A mixed effects logistic regression model was used to jointly analyse the observed primary outcomes. This approach assumes that, given their ANC attendances (recorded for nearly all women), whether a woman did or didn’t bring their clinic book for extraction of the other outcomes was unrelated to the values of those outcomes.”. See also comment 26
37	Page 8, lines 3-8 – the sentences may be refined to articulate more clearly in a lucid language about the random effects at individual and clinic level.	"For the clinic-level random effects, an unstructured covariance matrix was used with random effect terms for (1) ANC visits, (2) delivery at a healthcare facility and (3) PNC and child immunisation visits." reworded to "Clinic-level random effect terms were defined for (1) ANC visits, (2) delivery at a healthcare facility and (3) PNC and child immunisation visits, with unrestricted correlations between these." to improve communication of the model structure. However, we feel that the description of random effects at the level of each individual woman is appropriate as written.
38	“The secondary outcomes of counts of ANC clinic, PNC clinic and child immunisation visits were analysed using a multivariate Poisson mixed effects models” – since these events (visits) are not rare events, what was the rationale of using Poisson model!	Poisson regression models can be used to estimate the relative risk for binary outcomes (taking a value of 0 or 1) when an event is rare. However, we have used a Poisson regression model for observed count data (taking non-negative integer values). This is a natural choice of analysis model for such data, and

		the 'rare event' assumption is not required.
39	"All analyses were adjusted by the baseline maternal parity ('0' vs '≥1'), by the presence of any maternal medical conditions leading to classification of the pregnancy as high-risk (HIV with or without ART, diabetes, hypertension, malaria, each coded with separate indicator variables) and by the clinic-level randomisation stratification variable of sub-county." – it is important to mention what criteria were followed for selection of variables for adjustment.	This has now been described in the 'Statistical analysis' subsection of the Methods: "The analysis followed a pre-specified statistical analysis plan, which was finalised after data collection but prior to any unblinded analysis. Selection of maternal characteristics as adjustment variables was based on their inclusion in the core enrolment dataset, the associated absence of missing data for these items and their potential to predict the outcomes of interest. Sub-county was included as an adjustment variable because of its use as a stratification factor in the randomisation process for the study."

VERSION 2 – REVIEW

REVIEWER	Margaret McConnell Harvard University T H Chan School of Public Health, Global Health and Population
REVIEW RETURNED	28-Nov-2021
GENERAL COMMENTS	Thank you for the careful responses to reviewers' comments. I can't tell from the current manuscript whether the CONSORT diagram indicating eligibility and enrollment is included in the main manuscript. I think it should be as it is critical for understanding results.